# Colored sticky traps for monitoring phytophagous thrips (Thysanoptera) in mango agroecosystems, and their impact on beneficial insects

Lucia Carrillo-Arámbula[1], Francisco Infante[1]*, Adriano Cavalleri[2], Jaime Gómez[1], José A. Ortiz[1], Ben G. Fanson[3], Francisco J. González[4]

**1** El Colegio de la Frontera Sur (ECOSUR), Tapachula, Chiapas, México, **2** Universidade Federal do Rio Grande, Rio Grande, RS, Brazil, **3** Department of Environment, Land, Water and Planning, Arthur Rylah Institute for Environmental Research, Heidelberg, Victoria, Australia, **4** Universidad Autónoma de San Luis Potosí, San Luis Potosí, SLP, México

* finfante@ecosur.mx

**Data Availability Statement:** All relevant data are within the manuscript and its Supporting Information files.

## Abstract

The capture efficiency of six colored sticky traps (blue, green, orange, purple, white, and yellow) was tested in mango agroecosystems of Mexico with the purpose to: (i) document the diversity of Thysanoptera; (ii) determine the attraction of phytophagous thrips; (iii) assess the impact of these traps on beneficial insects; and (iv) assess the relationship between the density of *Frankliniella* thrips captured on traps and those found in the inflorescences. The use of colored sticky traps has revealed a great diversity of thrips and beneficial insects in the mango agroecosystem. A total of 16,441 thrips were caught on sticky traps throughout the sampling period, of which 16,251 (98.8%) were thrips adults and 190 (1.2%) larvae. Forty one species of thrips were collected either from sticky traps or from inflorescences. Of these, 32 species feed either on leaves or flowers. *Frankliniella cephalica*, *F. gardeniae* and *F. invasor*, were the most abundant species. *Scirtothrips citri* and *S. manihoti* were also captured among other phytophagous thrips. The white trap captured significantly more *Frankliniella* species and also had the smallest capture of beneficial insects. Yellow traps were the most attractive for *Scirtothrips* species, with low detrimental effects on insect pollinators, although high impact on natural enemies. Thrips species captured on sticky traps showed a low and non-significantly correlation with respect to the density of thrips in mango inflorescences. Although sticky traps did not predict the density of *Frankliniella* populations in mango inflorescences, the study represents a substantial progress in the use of color traps in mango agroecosystems. Colored sticky traps would be a good option for monitoring mango thrips to detect them at earlier stages of infestation to implement management tactics and avoid the building-up of thrips populations.

**Funding:** The authors received no specific funding for this work.

**Competing interests:** The authors have declared that no competing interests exist.

## Introduction

Mango (*Mangifera indica* L.) is one of the most important tropical fruits and is cultivated in more than one hundred countries [1]. First introduced into Mexico from The Philippines before 1779 [2], this crop flourished throughout the country, resulting in an important cash crop. Mexico is the largest exporter of mango worldwide. In 2020, approximately 465,000 tons were exported, representing 21% of the global export trading [3].

Ataulfo is a Mexican mango of great popularity in the international markets and one of the finest cultivars exported by this country [4, 5]. This cultivar originated in southern Mexico by the middle of the last century [6, 7], and since then, it has gradually become the predominant cultivar around the country [8], displacing the traditional Floridian mango cultivars (Haden, Keitt, Kent, Tommy Atkins, etc.).

In the State of Chiapas, Mexico, mango is one of the most important crops with approximately 39,000 ha, from which 85% are planted with Ataulfo [8]. During mango flowering, numerous species of insects invade orchards to feed and reproduce on inflorescences. Thrips are highly attracted to the cultivar Ataulfo. Rocha *et al*. [9] reported 15 thrips species in Chiapas, with seven *Frankliniella* spp. among the most abundant. Some *Frankliniella* species are polyphagous and well-known as pests of mango inflorescences [10–12]. Due to their opportunistic habits, they feed on the floral nectaries and anthers, ovipositing the rachis of panicles and flowers [13, 14]. *Frankliniella* larvae and adults damage plants by puncturing and then sucking the cellular contents [15]. Other pestiferous thrips are also frequently collected in mango flowers, i.e., *Scirtothrips* spp., although species in this genus are most common on young leaves [16] and small mango fruits.

Damage by thrips has been associated with the decline of mango yields in Chiapas [17]. Inflorescences heavily infested with thrips dry up, flowers drop off prematurely and fail to set fruit. Earlier studies have recorded a mean of 867 *Frankliniella* thrips (larvae and adults) per inflorescence throughout the flowering period [9]. Although the economic threshold for thrips damage has not been established, preliminary studies mentioned that inflorescences with more than 95% open flowers can tolerate up to 1,109 thrips per inflorescence without yield losses [18]. Because of their small size, mango thrips are challenging to be sampled and counted. Usually, thrips management is carried out by spraying synthetic insecticides regardless of the thrips density population [18]. In other agroecosystems, the use of colored sticky traps has been suggested for monitoring purposes to estimate thrips populations due to their low cost and rapid implementation [19–21].

Few studies have reported the attraction of thrips to different colored sticky traps for monitoring in mango plantations [22, 23]. Thrips captures on sticky traps have also been used to estimate the population density in inflorescences [24, 25], and as an early warning for thrips infestation [26]. However, these studies have been conducted in mango agroecosystems with different thrips species to those present in Mexico. The single study that assessed sticky traps on Ataulfo mango in Mexico was performed by Virgen-Sánchez *et al*. [27]. They found that purple traps were more attractive for thrips than the blue and yellow ones. However, that study evaluated three colors only and did not identify the thrips species caught on traps. The impact on non-target insects was also overlooked.

Plans for thrips management involve developing a sampling tool for monitoring thrips and implementing control measures at the right time to control them. In this sense, the present study aimed to: 1) document the diversity of Thysanoptera fauna in mango agroecosystems; 2) determine the attractiveness of phytophagous thrips, such as the *Frankliniella* and *Scirtothrips* species, towards several colored sticky traps; 3) assess the impact of these traps on beneficial insects, with special emphasis in natural enemies (parasitoids and predators) and pollinators;

and 4) assess the relationship between the density of *Frankliniella* thrips captured on colored sticky traps and those found in the inflorescences.

## Materials and methods

### Site description

The experiment was conducted in the Ataulfo mango orchard El Vergel (N14˚42'04"; W92˚ 19'05"; 15 meters above sea level; m.a.s.l.), near Tapachula, Chiapas, Mexico. This commercial orchard has an extension of 70 ha. Mango trees were planted at a density of 68 trees/ha, and the production is usually exported. Because of the quantity of agrochemicals used, it is considered that the orchard receives intensive agronomic management. Each year, mango flowering is induced by the middle of November by spraying potassium nitrate ($KNO_3$) on trees. After flowering, farmers use a micro-spray irrigation system that constantly works during the fructification period. Several insecticides and fungicides are sprayed to control pests and diseases. Weed control is carried out mechanically and complemented with amine-based herbicides. Urea and potassium phosphate are used regularly as fertilizers.

### Colored sticky trap preparation

Double-sided colored sticky traps used in this study were homemade constructed, using a 15 x 21 cm cardboard lined with colored self-adhesive paper (Office Depot de México, S.A. de C. V.). We selected the following colors: blue, green, orange, purple, white, and yellow. Separately, an acetate sheet same size of the cardboard was slightly coated on one side with tangle glue (The Tanglefoot Company, Marysville, Ohio, USA). Two acetates were fastened with plastic clips to each side of the colored cardboards to catch insects on both sides of the traps. The advantage of using acetates over the color cardboards is that traps did not need replacing after each sampling, as we only removed the acetates with the stuck insects.

### Experimental procedure

The experimental plot was an area of ca. 5.5 ha. We used a completely randomized blocks design to evaluate the thrips and beneficial insects' attraction to color traps in the mango agroecosystem. Traps were randomly deployed in a 2 x 1.75 m (height and wide, respectively) T-shape stake, similar to those used by Hoddle *et al*. [28]. Color traps were separated by approximately 5 cm each other on the T-stake (Fig 1). Each block was a T-shape stake with all six treatments (colors) in random order. During each sampling date, eight T-stakes were placed evenly spaced throughout the orchard. They were perpendicularly orientated to the row of trees to reduce the possibility that thrips dwelling inflorescences were pushed to the traps by the wind. Traps were maintained in the field for 72 h and then removed and taken to the laboratory to identify thrips and other insects. The same procedure was repeated every 10 days for a total of eight sampling dates along the flowering period. Experiments began on December 7, 2019, following the initiation of mango flowering, and finished on February 18, 2020. The average temperature at the experimental plot was 28± 9˚C and 75± 25% RH.

### Sample processing

Once in the laboratory, each acetate was removed from traps and superimposed on a millimeter paper as a background to facilitate insects' counting. The number of individuals caught on traps was counted with the aid of a dissecting microscope. In the case of thrips adults, a subsample based on the different morphotypes was randomly taken from each color to be mounted on slides, to corroborate the thrips identification at species level. For this, thrips were

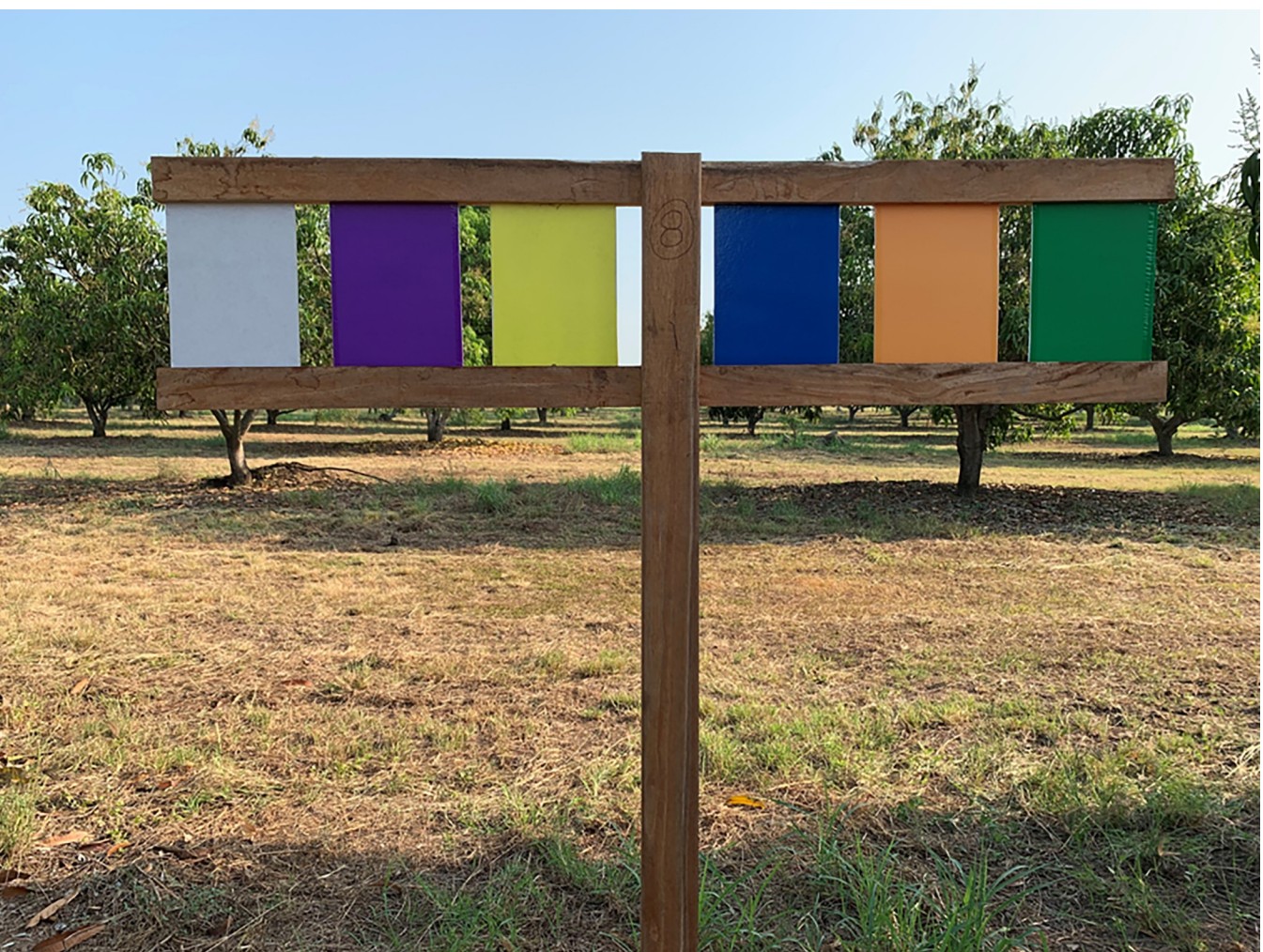

**Fig 1. The colored sticky traps.** Manner in which colored sticky traps were disposed in the mango orchard to attract thrips and other insects.

carefully detached by cutting the acetate into smaller pieces and placed in a Petri dish with white gasoline as a dissolvent for 15 minutes. Subsequently, thrips were transferred to Eppendorf vials having distilled water and shaken for three minutes to remove the remaining gasoline. Once clean, thrips were kept in vials with 70% ethanol for several weeks. Specimens were soaked in a 5% NaOH solution for four hours at room temperature, and the abdomen was punctured to expel the internal body content. Thrips were mounted on slides using Hoyer's medium and dried in an oven (45°C) for one week. Thrips were identified using specialized taxonomic keys [29–32]. Voucher specimens were deposited at the entomological collection of El Colegio de la Frontera Sur in Tapachula, Chiapas, Mexico. Although Canada balsam is the traditional and permanent mounting medium for thrips on slides, we preferred Hoyer' medium, since this method require shorter steps in preparing specimens and allow us to identified thrips adults rapidly. This last method has been recommended when there is a large number of individuals to be mounted [33], and specimens can be preserved without deterioration up to 20 years [34]. Thrips larvae caught in sticky traps were counted but not identified at species level.

The non-target insects captured on sticky traps were identified to the family level, following the taxonomic keys in Goulet and Huber [35]; Arnett and Thomas [36]; Arnett et al. [37];

Triplehorn and Johnson [38]; and Brown *et al.* [39]. Based on the scientific literature, the beneficial insects were separated in two ecological groups: parasitoids and predators [40, 41] and insect mango pollinators [42–45].

## Reflectance of colored sticky traps

Diffuse reflectance spectrophotometry, which measures the light reflected by objects as a function of wavelength, has been widely employed to evaluate the color of biological and artificial pigments [46]. For the colored sticky traps, their spectral reflectance (without the acetate and the glue) was determined by a visible-infrared spectrophotometer (USB4000-VIS-NIR, Ocean Optics, Orlando, FL) with an optical resolution of 1.5 nm (full width at half maximum). Even though our spectrometer provided readings down to 350nm the readings between 350 and 375nm were noisy and not reliable, therefore the reliable range of 375 to 1000nm was considered for this work (Fig 2). The illumination of sticky traps was performed with a tungsten-halogen light source (LS-1, Ocean Insight, Orlando, FL). Both the illumination and the reflected light were provided and collected respectively using a fiber-optic reflection probe (R200-7-VIS-NIR, Ocean Optics, Orlando, FL). Measurements were performed using a 1s integration time. The raw reflectance spectra were corrected to eliminate the dark current and normalized with the spectrum obtained from the light source reflected on a Teflon white standard reference [47].

## Sampling of mango inflorescences

At each sampling date of colored sticky traps, 20 mango inflorescences were collected by hand in the experimental plot. Ten of these inflorescences were gathered simultaneously when the

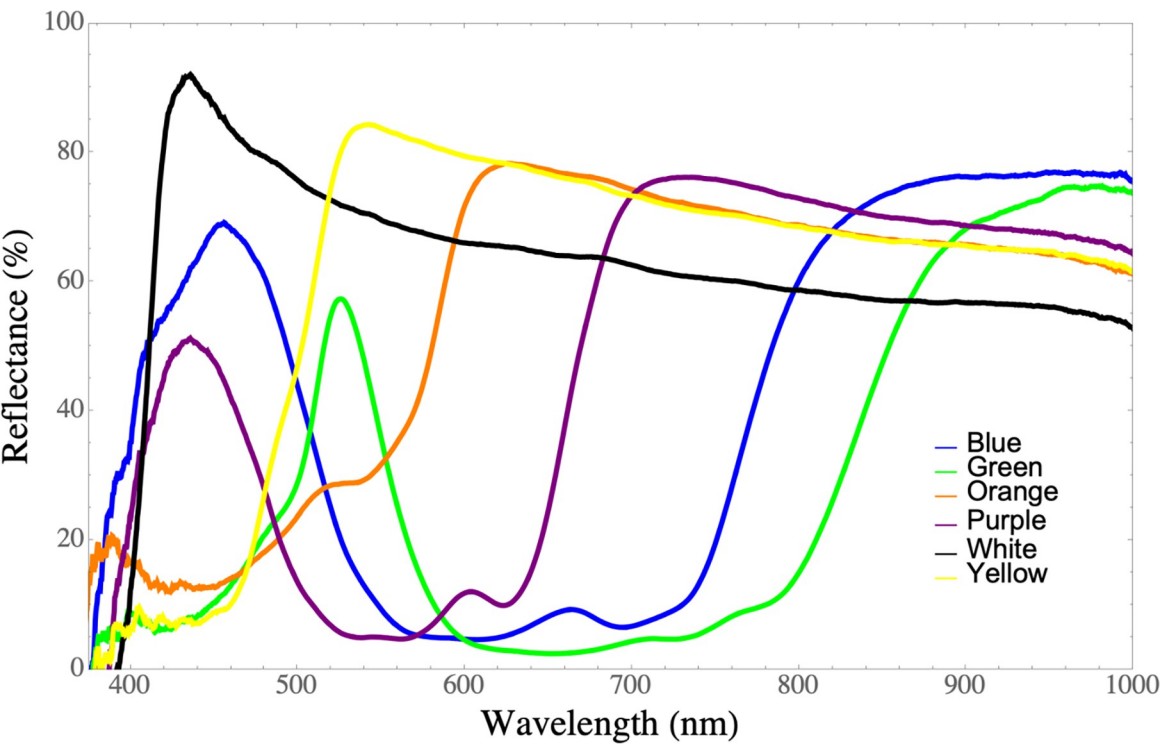

**Fig 2. Reflectance curves.** Spectral reflectance curves (%) of the six color traps used in the experiment to attract thrips adults in a mango orchard.

sticky traps were set up in the field, and the other 10 inflorescences were collected 72 hours later, i.e., when sticky traps were removed from the field. The same procedure was repeated every 10 days, for a total of seven sampling dates along the flowering period; the eighth planned sampling was not taken because there were no inflorescences in the field. We determined whether there was any relationship between the density of *Frankliniella* thrips captured on the sticky traps and the number of *Frankliniella* thrips existing in mango inflorescences.

During samplings, inflorescences were collected at random between 08:00 and 10:00 hours from different trees, and about 2 m above the ground. They were placed individually in plastic bags and kept in an icebox before being taken to the laboratory. Samples were processed by rinsing the bag and contents in 70% ethanol to kill insects. Subsequently, the bag's contents were shaken and sieved. Using different sieve size gradations, this procedure was repeated several times until insects detached from flowers. Insects were then collected from the ethanol solution and preserved in 70% ethanol. Thrips were separated and counted under a stereomicroscope; the beneficial insects were kept in vials for identification. Regarding the thrips identification, due to the high variability of the thrips numbers in inflorescences, we mounted subsamples of adult thrips on slides to be identified, according to the following quantities: samples containing up to 10 thrips, 100%; from 11 to 50 thrips, 50%; 51–100, 20%; 101–500, 7%; 501–1000, 3%; 1001–2000, 2%; and 2000–5000, 1% individuals. In this manner, we mounted 3,015 specimens for identification. Thrips larvae were counted, but not identified to species level, since our main objective was to correlate the adult thrips from inflorescences to those found on sticky traps.

## Statistical analysis

To test the effect of trap color on catch rates for thrips and beneficial species, we ran five separate analyses for the following groups: *Frankliniella*, *Scirtothrips*, other thrips, natural enemies, and insect pollinators. We performed a generalized linear mixed model (GLMM) where a negative binomial distribution was assessed and confirmed. Graphical assessment of Pearson residuals was used to assess the model assumptions. For the fixed effects, we included color treatment and sampling date, as well as the interaction. The block of T-stakes was included as random effects. If the interaction was not significant (log-likelihood ratio test), we removed the interaction from the model. For any significant treatment effect, a pairwise comparison with a Tukey correction was performed. For significant treatment by sampling date, we ran pairwise treatment comparison sliced by date, again with a Tukey correction. Finally, we also compared colors with peak reflectance in the short wavelengths (blue, purple, white) to colors with longer wavelengths (green, orange, yellow) using a linear contrast.

We used correlation analyses to assess the relationship between the density of *Frankliniella* thrips captured on sticky traps and those found in the inflorescences. As the experimental design was balanced and replicates were not paired, we averaged across all replicates to get a mean catch rate for each sampling date for the trap dataset of each color. Similarly, we obtained the mean of thrips across all replicates for the inflorescence dataset to get a single total catch rate for adults for each sampling date. We then used the square root transformation to normalize the data and ran Pearson's correlation analysis between thrips on sticky traps and inflorescences. All analyses were conducted using the R software [48].

## Results

### Thysanoptera diversity in mango agroecosystems

A total of 16,441 thrips were caught on sticky traps throughout the sampling period, of which 16,251 (98.8%) were thrips adults and 190 (1.2%) larvae (S1 Table). The number of thrips in

the inflorescences was much higher than those caught on traps. A total of 439,352 individuals were collected in the inflorescences. From these, 97,294 (22.1%) were adults, and 342,058 (77.9%) larvae (S2 Table).

Samplings revealed the presence of 41 thrips species in the mango agroecosystem. The highest number of species (37) was collected with sticky traps, while 13 species were collected from inflorescences. Only nine species were recorded in both methods of capture (Table 1). According to the feeding habits reported for these species, five are known to feed on fungi, four are predators of thrips as well as other small arthropods, and 32 feed either on leaves or flowers.

Most adult thrips captured on sticky traps belonged to the genus *Frankliniella*, representing 88% of the total individuals caught, followed by 9% for *Scirtothrips* species and 3% for species of other genera. Likewise, the species of *Frankliniella* were the most abundant individuals in inflorescences, comprising 99% of the adults identified in the subsamples. From them, *F. invasor* was the dominant species having on average 71.9% of individuals, followed by *F. gardeniae* (17.3%), *F. cephalica* (9.5%), and other *Frankliniella* species (<0.1%) (Table 2).

## Attraction of thrips to colored sticky traps

Considering the eight samplings as a whole, the trap color affected catch rates for *Frankliniella*, *Scirtothrips*, and other species of thrips too. For *Frankliniella* thrips, the white trap caught significantly more individuals than any other color tested ($\chi^2$ = 180.2; df = 5; P < 0.001; Fig 3A). In general, *Frankliniella* thrips were more attracted to colors with reflectance peaks between 400 to 460 nm. Average capture rates for shorter wavelength colors were 76% (p < 0.001; 95% CI: 59% - 94%) higher than the longer wavelength colors. In contrast to *Frankliniella*, the yellow trap had the highest catch rate for *Scirtothrips* ($\chi^2$ = 560.8; df = 5; P < 0.001; Fig 3B). Species of *Scirtothrips* responded better to colors with reflectance peaks between 500 to 550 nm. On average, the longer wavelengths, such as, green, orange and yellow caught 349% more individuals (P < 0.001; 95%CI: 288% - 418%) than shorter wavelengths, such as, blue, purple, and white. For the other thrips, catch rates were very low and the significant effect of color was driven by the green traps having higher catch rates than white traps ($\chi^2$ = 15.8; df = 5; P < 0.007), but otherwise no other significant pairwise differences were present (Fig 3C). Comparison of long vs short wavelengths found a 19% decrease (P = 0.03; 95%CI: 2% - 33%) in capture rates for short wavelength traps compared to long wavelength traps.

When comparing the interaction between treatments (color) with respect to the dates of sampling, only *Frankliniella* had a significant effect on the rate of captures ($\chi^2$ = 147; df = 35; P < 0.001), indicating that catches by the colored traps were different in the eight samplings (Fig 4). For instance, in the first date, white traps caught more than two times the number of *Frankliniella* thrips compared with the other traps. However, in the following samplings, no significant pairwise differences were found in the *Frankliniella* catches between the white and blue traps in six out of eight samplings. Similarly, there were no differences in the *Frankliniella* catches between the white and purple traps in five of the eight samplings. There was no interaction for *Scirtothrips* spp. ($\chi^2$ = 37.7; df = 35; P = 0.35) or the other species of thrips captured ($\chi^2$ = 33.2; df = 35; P = 0.56).

## Impact of sticky traps on beneficial insects

Color treatments had a significant effect on the catch rates of natural enemies ($\chi^2$ = 48.8; df = 5; P < 0.001; Fig 5A) but did not significantly affect pollinators' catches ($\chi^2$ = 5.7; df = 5; P = 0.33; Fig 5B). The yellow, green, and orange traps caught the highest numbers of natural enemies and there was no significant difference among them. White, blue, and purple traps were less effective in capturing natural enemies. The interaction between treatment and

**Table 1. Diversity of thrips species inhabiting Ataulfo mango agroecosystems.**

| Species | Sticky traps | Inflorescences |
|---|:---:|:---:|
| **Terebrantia** | | |
| Aeolothripidae | | |
| *Ambaeolothrips romanruizi* (Ruiz-De la Cruz et al. 2013) | ✓ | ✓ |
| *Stomatothrips flavus* Hood, 1912 | ✓ | |
| Heterothripidae | | |
| *Heterothrips decacornis* Crawford DL, 1909 | ✓ | |
| Thripidae | | |
| *Arorathrips mexicanus* (Crawford DL, 1909) | ✓ | |
| *A. spiniceps* (Hood, 1915) | ✓ | |
| *Bregmatothrips venustus* Hood, 1912 | ✓ | |
| *Caliothrips phaseoli* (Hood, 1912) | ✓ | |
| *Chaetanaphothrips leeuweni* (Karny, 1914) | ✓ | |
| *Frankliniella borinquen* Hood, 1942 | | ✓ |
| *F. cephalica* (Crawford DL, 1910) | ✓ | ✓ |
| *F. gardeniae* (Moulton, 1948) | ✓ | ✓ |
| *F. insularis* (Franklin, 1908) | ✓ | |
| *F. invasor* Sakimura, 1972 | ✓ | ✓ |
| *F. parvula* Hood, 1925 | | ✓ |
| *Halmathrips citricinctus* Hood, 1936 | ✓ | |
| *H. tricinctus* Stannard, 1953 | ✓ | |
| *Heliothrips haemorrhoidalis* (Bouché, 1833) | ✓ | ✓ |
| *Hydatothrips nr. gliricidiae* Mound & Marullo, 1996 | ✓ | |
| *H. sternalis* (Hood, 1935) | ✓ | |
| *Leucothrips furcatus* Hood, 1931 | ✓ | |
| *Macrophthalmothrips helenae* Hood, 1934 | ✓ | |
| *Microcephalothrips abdominalis* (Crawford DL, 1910) | ✓ | |
| *Neohydatothrips gracilipes* (Hood, 1924) | ✓ | |
| *N. inversus* Hood, 1928 | ✓ | |
| *Plesiothrips perplexus* (Beach, 1896) | ✓ | |
| *Salpingothrips minimus* Hood, 1935 | ✓ | |
| *Scirtothrips citri* (Moulton, 1909) | ✓ | ✓ |
| *S. manihoti* Bondar, 1924 | ✓ | ✓ |
| *Scirtothrips* sp. | | ✓ |
| *Scolothrips pallidus* (Beach, 1896) | ✓ | |
| *Thrips* sp. | ✓ | |
| **Tubulifera** | | |
| Phlaeothripidae | | |
| *Allothrips megacephalus* Hood, 1908 | ✓ | |
| *Androthrips ramachandrai* Karny, 1926 | | ✓ |
| *Diceratothrips bicornis* Bagnall, 1908 | ✓ | |
| *Gastrothrips* nr. *fulvicauda* Hood, 1937 | ✓ | |
| *Gynaikothrips uzeli* (Zimmermann, 1900) | ✓ | ✓ |
| *Karnyothrips texensis* (Hood, 1940) | ✓ | ✓ |
| *Liothrips jatrophae* (Moulton, 1929) | ✓ | |
| *L.* nr. *tabascensis* Johansen, 1976 | ✓ | |
| *Liothrips* sp. | ✓ | |
| *Strepterothrips floridanus* (Hood, 1938) | ✓ | |

**Table 2. Thrips from mango inflorescences.**

| Species | Samplings | | | | | | |
|---|---|---|---|---|---|---|---|
| | 1 | 2 | 3 | 4 | 5 | 6 | 7 |
| **Terebrantia** | | | | | | | |
| Aeolothripidae | | | | | | | |
| *Ambaeolothrips romanruizi* | 0 | 0.2 | 0.6 | 0.2 | 0.2 | 0 | 0 |
| Thripidae | | | | | | | |
| *Frankliniella borinquen* | 0 | 0.2 | 0 | 0 | 0 | 0 | 0 |
| *F. cephalica* | 3.3 | 13.6 | 3.2 | 10.4 | 9.3 | 6.1 | 21.0 |
| *F. gardeniae* | 11.2 | 27.0 | 34.6 | 14.7 | 7.6 | 8.6 | 17.6 |
| *F. invasor* | 84.7 | 58.5 | 61.1 | 73.3 | 81.3 | 85.1 | 59.7 |
| *F. parvula* | 0 | 0 | 0 | 0 | 0.2 | 0 | 0 |
| *Heliothrips hemorroidalis* | 0.2 | 0 | 0 | 0 | 0 | 0 | 0 |
| *Scirtothrips citri* | 0.4 | 0.2 | 0.4 | 1.4 | 0.4 | 0.2 | 0.8 |
| *S. manihoti* | 0.2 | 0 | 0 | 0 | 0 | 0 | 0 |
| *S. nr. dorsalis* | 0 | 0.2 | 0 | 0 | 0 | 0 | 0 |
| **Tubulifera** | | | | | | | |
| Phlaeothripidae | | | | | | | |
| *Androthrips ramachandrai* | 0 | 0 | 0 | 0 | 0.2 | 0 | 0 |
| *Gynaikothrips uzeli* | 0 | 0 | 0 | 0 | 0.4 | 0 | 0 |
| *Karnyothrips texensis* | 0 | 0 | 0 | 0 | 0.2 | 0 | 0.8 |

Species composition of thrips adults collected from mango inflorescences in seven samplings throughout the flowering period of Ataulfo mango. Figures in each column represent the percentage of thrips species in a subsample that was mounted on slides (n = 3,015 specimens).

*Sampling eight was not carried out because there were no mango inflorescences in the field.

sampling date showed no significant differences in the catch rates of either natural enemies ($\chi^2$ = 20; df = 35; P = 0.98) or insect pollinators ($\chi^2$ = 17.9; df = 35; P = 0.99).

Sticky traps captured 5,136 beneficial arthropods throughout the sampling period. A total of 83% of species captured were insect natural enemies and 17% were insect mango

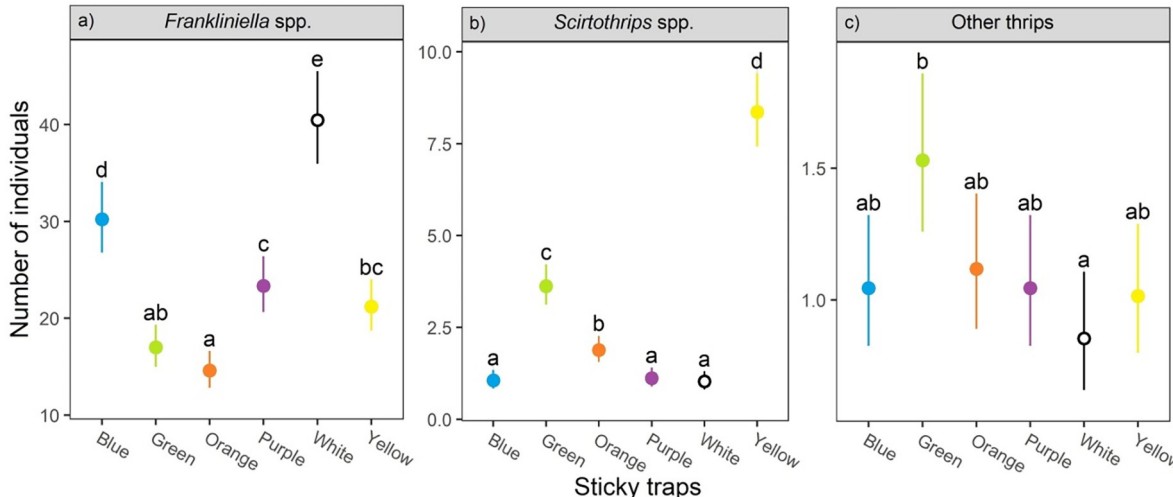

**Fig 3.** Attraction of thrips adults to colored sticky traps: a) *Frankliniella* spp., b) *Scirtothrips* spp., and c) other thrips. Each point shows the mean catches of eight replicates in eight sampling dates, and the estimate marginal mean and error bars at 95% CI. Different letters indicate Tukey significant differences (p<0.05).

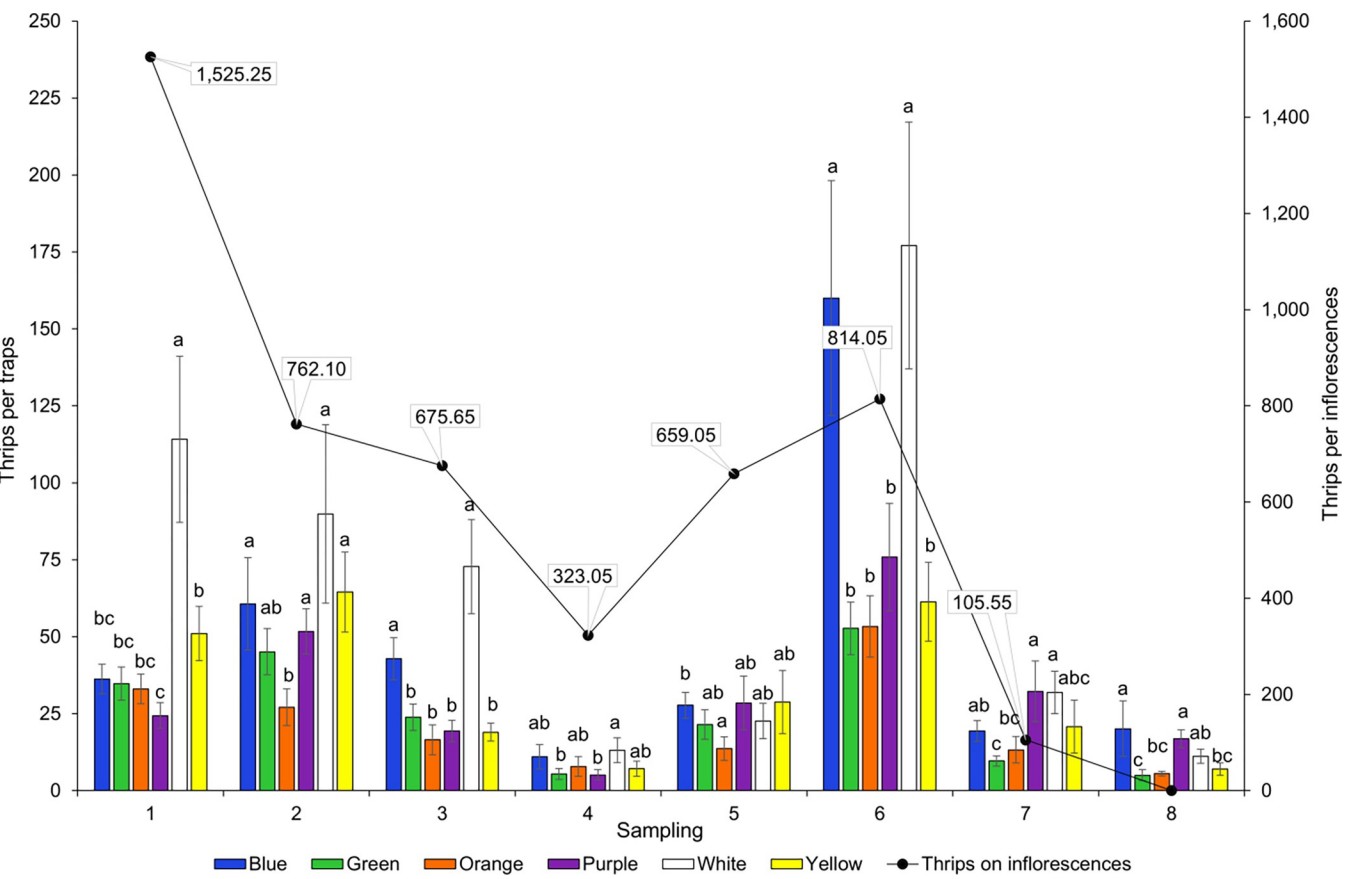

**Fig 4. Attraction of thrips adults of the genus *Frankliniella* to colored sticky traps.** Each bar represents the mean catches (±SE) of eight replicates to six colors in eight sampling dates (axis-$y_1$). The line shows the mean (±SE) thrips adults of *Frankliniella* in 20 inflorescences in seven sampling dates (axis-$y_2$). Bars capped with the same letter within a sampling date are not significant different (p>0.05) (untransformed data).

pollinators. The natural enemies included 41 insect families in six orders and members of the order Araneae (Fig 5C). Hymenoptera and Coleoptera were the insect orders more diverse with 19 and 9 families, respectively. Araneae comprised the highest number of individuals trapped, followed by hymenopteran Scelionidae, Mymaridae, Encyrtidae and Aphelinidae (S3 Table). Regarding insect pollinators, 12 families in the orders Coleoptera, Diptera, and Hymenoptera, were identified (Fig 5D). Sciaridae, Milichiidae, Formicidae, and Chloropidae were the families with the highest number of individuals (S4 Table).

## Relationship between *Frankliniella* thrips caught on traps *versus* inflorescences

We found positive although non-significant correlations between the numbers of *Frankliniella* caught on colored sticky traps and those collected from inflorescences (Fig 6). The green trap had the highest correlation values (*r* = 0.73, P = 0.06) in comparison to the other color traps. The overall catches of *Frankliniella* thrips on traps and inflorescences along the sampling dates can be seen in Fig 4. In general, the highest captures of thrips on sticky traps coincided with the high thrips density in the inflorescences. Thrips abundance was highly variable throughout the study, either in traps or inflorescences. In the latter, the overall average was 3,138 thrips per inflorescence (695 adults and 2,443 larvae; S5 Table).

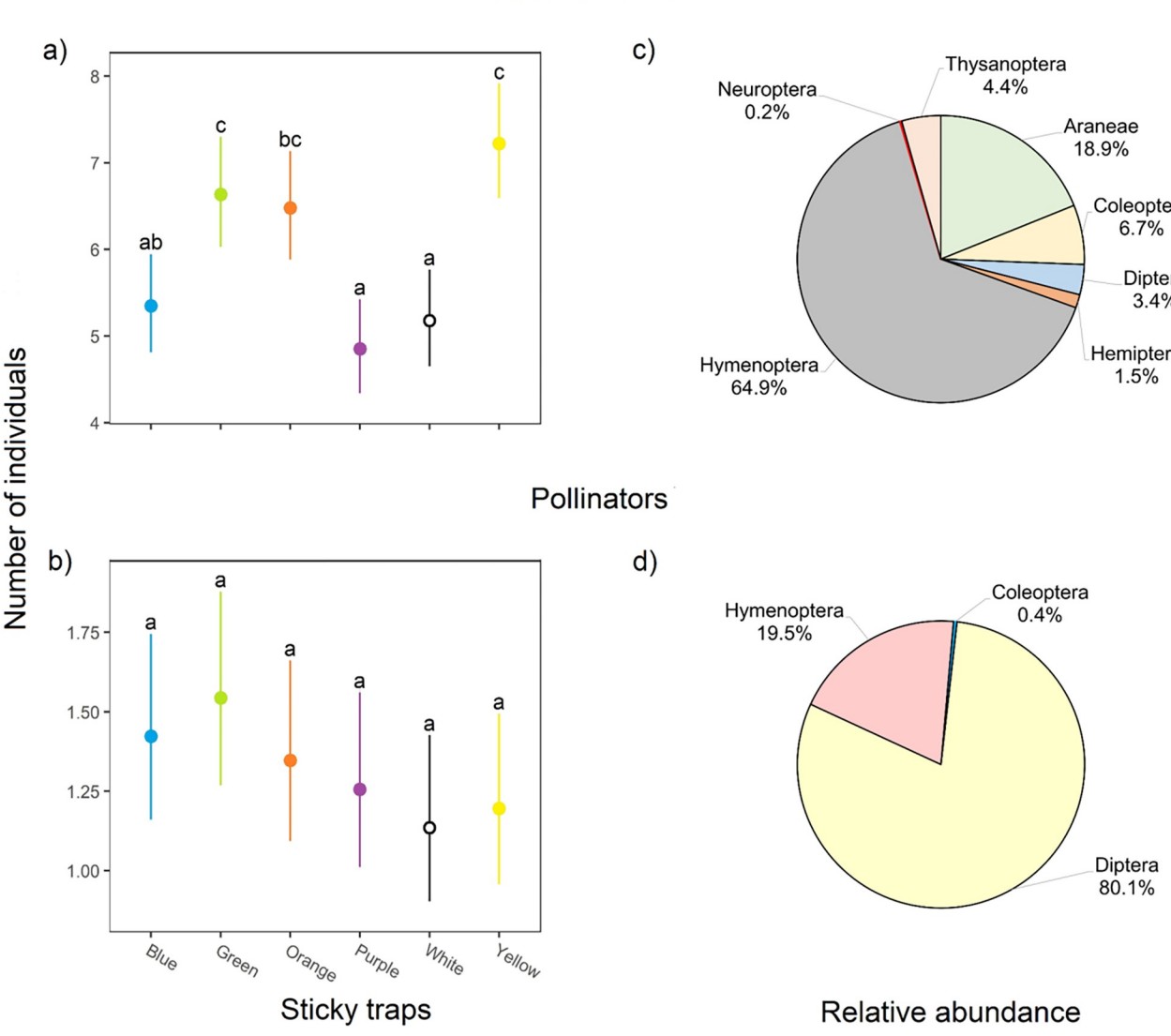

**Fig 5. Attraction of beneficial insects to colored sticky traps during the Ataulfo mango flowering.** Each point shows the mean catches of eight replicates in eight sampling dates, and the estimate marginal mean and error bars at 95% CI for natural enemies (a) and pollinators (b). Different letters indicate Tukey significant differences (p<0.05). Relative abundances (%) of insect orders (plus Aranea) for natural enemies (c) and pollinators (d) are presented on the right side.

## Discussion

### Diversity of Thysanoptera in mango agroecosystems

The use of colored sticky traps has revealed a great diversity of thrips and beneficial insects in the Ataulfo mango agroecosystem. With regard to thrips, after collecting more than 455 thousand individuals, either with traps or directly from mango inflorescences, we identified 41 species in 28 genera and four families. We are not aware of a similar study in this crop with such an extensive sampling effort.

Despite the high species richness, we assumed that most thrips species were not feeding on mango flowers, as numerous thrips captured on traps were not captured in the inflorescences. These species were presumably living on grasses and herbaceous plants near mango trees or

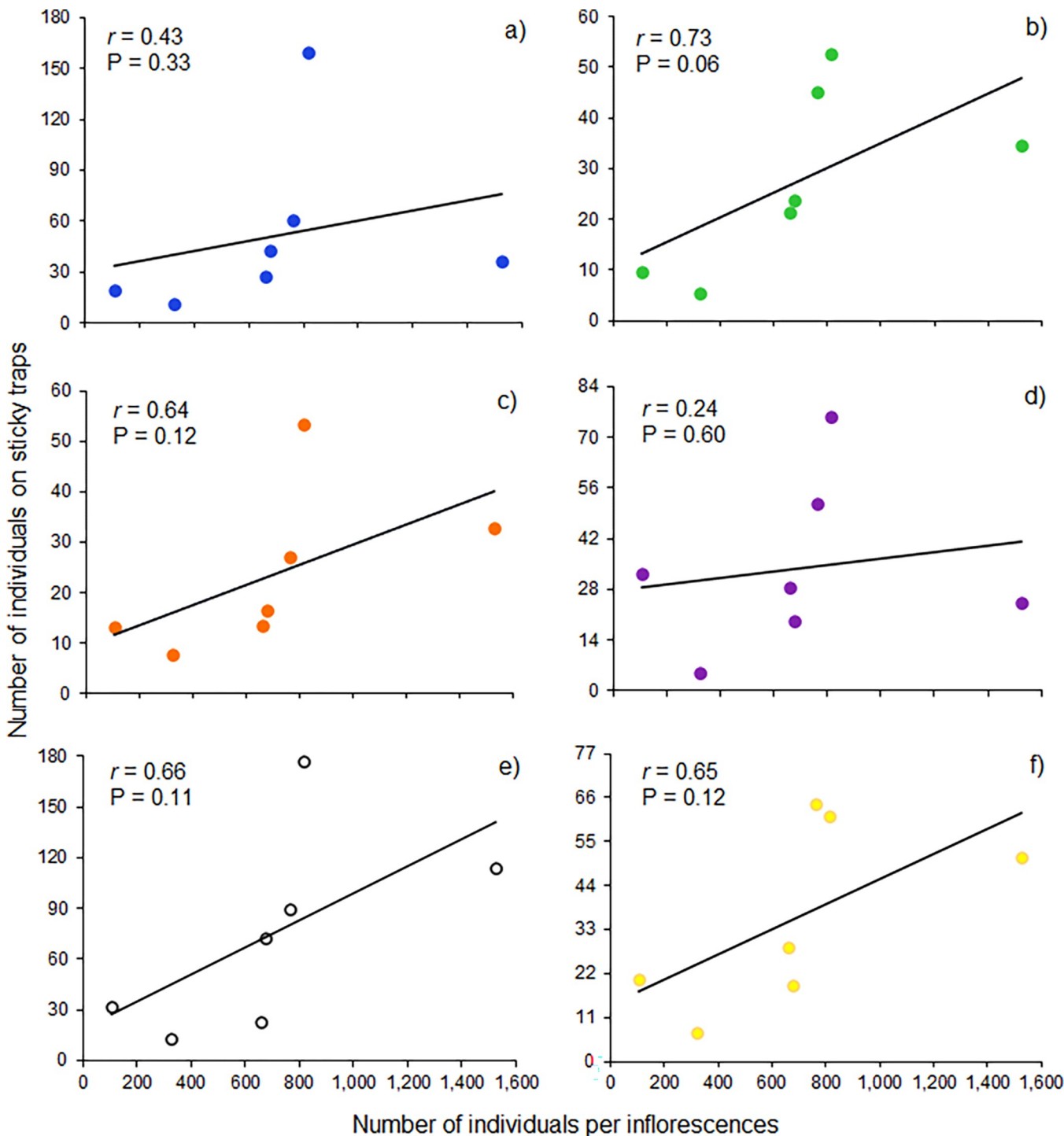

**Fig 6. Thrips caught on traps *versus* inflorescences.** Relationship between the average number of *Frankliniella* species captured on colored sticky traps (mean catches for each sampling date) and the average number of *Frankliniella* in the inflorescences (mean catches in 20 inflorescences): a) blue, b) green, c) orange, d) purple, e) white and f) yellow.

came from the surrounding vegetation of the mango orchard. It is well known that weeds are usually hosts of numerous species of thrips and other insects, serving as a temporary refuge [49, 50]. For example, species in the genera *Arorathrips*, *Bregmatothrips* and *Plesiothrips*, are

known to breed only on grasses (Poaceae) [51]. Some recorded species are known to be highly specific to certain plants, as *Gynaikothrips uzeli* Zimmermann, which feeds on leaves of ornamental figs [52], and *Microcephalothrips abdominalis* (Crawford DL), known as the 'composite thrips' due to its preference for Asteraceae [53]. Predatory species of thrips in the genera *Karnyothrips*, *Scolothrips* and *Stomatothrips*, are common on grasses and herbaceous plants, where they feed on mites and other thrips [54, 55], whereas *Androthrips ramachandrai* Karny is known as a predator of *Gynaikothrips* spp. [56]. Species of the genera *Allothrips*, *Diceratothrips*, *Gastrothrips*, *Macrophthalmothrips*, and *Strepterothrips* feed on fungi and are usually collected from leaf litter and dead branches of trees [57–61]. Although colored sticky traps have been traditionally used for monitoring insect populations, the high numbers of thrips and other non-target species recorded in the present study, suggest that they can also be useful for biodiversity studies in mango agroecosystems. This approach has already been used in olive agroecosystems, to study arthropod aggregation, richness, diversity, and distribution [62].

Our results on thrips diversity are in agreement with other studies. Most of the 14 species of thrips that we collected in the inflorescences had already been recorded from Ataulfo mango orchards in Chiapas [9, 16, 18, 63]. However, there are other thrips collected on traps that have never been recorded either in mango or other plants in Mexico. As far as we know, this is the first record of the following species: *Chaetanaphothrips leeuweni* (Karny), *Halmathrips citricinctus* Hood, *H. tricinctus* Stannard, and *Scirtothrips manihoti* Hood. The species *C. leeuweni*, also known as the banana rust thrips, feeds and breeds on banana leaves, on which sometimes is a pest [29, 64]. Species of *Halmathrips* are generally feeding on forest trees' leaves [29]. *Scirtothrips manihoti* has been typically found damaging cassava leaves in several countries of Central and South America, and apparently its presence is strictly limited to this plant [29].

## Attraction of phytophagous thrips to colored sticky traps

Our findings of the *Frankliniella* attraction towards colored sticky traps suggest that although thrips catches were highly variable throughout the flowering period of mango, the white trap captured more adults than any other color in the eight samplings altogether. However, when considering the interaction between treatments (colors) and samplings dates, the white and blue traps captured statistically the same numbers of *Frankliniella* thrips in six out of eight samplings. Likewise, thrips captures by the white and purple traps were not significantly different in five of those samplings. In this manner, the sequence of attraction would be white->blue>purple, with slight differences among them. Green and orange traps appeared to be the least attractive colors to *Frankliniella*. Numerically, most of the *Frankliniella* captures on sticky traps and inflorescences consisted of three main species: *F. cephalica*, *F. gardeniae*, and *F. invasor*. It is important to emphasize that we measured the color preference for all *Frankliniella* species together, and it is possible that the variation found in the species composition of these three species throughout the mango flowering (Table 2), influenced the *Frankliniella* catch rates on different traps (Fig 4), since each particular species should have its own color preference. It is well known that insect catches in colored sticky traps depend on many factors, where insect behavior and other biological traits are involved [19]. In this sense, different species of thrips are attracted to different color traps [20]. Future studies should test the specific response of each *Frankliniella* species to determine color preferences. However, in this study we treated the *Frankliniella* species as a whole for practical purposes, considering they are phytophagous on mango flowers [9, 65] and with purportedly small differences in their biology.

Although our results of thrips catches are different to those reported by Virgen Sánchez *et al.* [27], who mentioned the purple trap as the most attractive for Ataulfo mango thrips, it has to be considered that they did not include the white color trap in experiments. Moreover,

in that study, thrips caught on sticky traps were not discriminated by species or genera, i.e., all thrips species were included in the color response, whereas we excluded all genera different to *Frankliniella* or *Scirtothrips*. Thus, differences between the two studies could be explained by the methods used.

The color preference by *Frankliniella* species has been studied in several crops. Responses are varied, but generally, blue, and white colors have been considered the most attractive. The white color was reported the best capturing *Frankliniella intonsa* (Trybom) in cowpea greenhouses [66], *Frankliniella occidentalis* Pergrande in avocado orchards [30], and *Frankliniella bispinosa* Morgan, in citrus groves [67]. Conversely, blue traps were more attractive to *Frankliniella schultzei* (Trybom) in bean fields [68], and *F. intonsa* in mango orchards of China [23].

This study confirmed that yellow traps were consistently the most attractive to *Scirtothrips* species throughout all samplings. In contrast to *Frankliniella*, white, blue, and purple traps were the less attractive for *Scirtothrips*. The color yellow is considered to be universally attractive to all foliage-seeking insects and has been extensively used in capturing important agricultural insect pests of different orders [69–71]. Its major disadvantage is that also attract many beneficial insects [72]. Our study is in line with previous findings that mentioned the yellow color as the most effective in capturing *Scirtothrips* species in mango [22, 25], avocado [28, 73], and pepper [74].

## Impact of traps on beneficial insects

One of the undesirable effects of sticky traps in agriculture is their impact on non-target organisms. In our experiments, over five thousand of beneficial insects were captured by traps during samplings. Natural enemies were by far, more numerous than insect pollinators. Green, orange, and yellow traps caught the higher numbers of natural enemies, whereas the blue, purple, and white had a lesser effect. Yellow sticky traps are known to trap high numbers of natural enemies of Diptera and Hymenoptera [75]. An evaluation of sticky traps for monitoring thrips in a cowpea crop, found that beneficial insects were caught 1.7 times more on yellow than on blue traps [20]. Our results suggest the monitoring of *Frankliniella* thrips using the white trap that also has the least detrimental effect on natural enemies.

Surprisingly, the impact of traps in capturing insect pollinators was similar for all colors tested. From the 12 insect families captured on sticky traps in the orders Coleoptera, Diptera and Hymenoptera, there were no catches for species in the families Calliphoridae, Muscidae, Sarcophagidae and Syrphidae. These families are very important in mango orchards because most insect pollinators of mango belong to such groups [76, 77]. It was established that green sticky traps captured on average the highest numbers of insect mango pollinators with 3.2 individuals/trap per sampling date; whereas white traps captured the lowest numbers with 1.5 individuals/trap. According to these low numbers, the use of color sticky traps in mango orchards would not be detrimental for insect mango pollinators.

## Thrips catches on the sticky traps *versus* thrips in the inflorescences

Although this study represents a substantial progress in the use of color traps in mango agroecosystems, sticky traps catches did not predict the density of *Frankliniella* populations in mango inflorescences. The lack of a significant correlation could be explained by the high variation in thrips densities on traps and inflorescences over time [78]. Further studies are required to find this relationship that should focus on improving the sampling methods to reduce the sampling variability to increase the degree of correlation. Estimation of *Frankliniella* thrips densities in mango inflorescences based on sticky traps would be of great importance in sampling thrips for management purposes.

A comparative study was conducted in Malaysia on the main pest species of mango flowers in that country: *Frankliniella schultzei* (Trybom), *Megalurothrips usitatus* Bagnal, *Scirtothrips dorsalis* Hood, and *Thrips hawaiiensis* (Morgan). It was determined that the number of thrips inhabiting mango inflorescences was highly correlated with the number of thrips caught on yellow sticky traps [25]. We think that these high correlations were significant because of the low variances found in samples. In our study, *Frankliniella* adults caught in sticky traps and inflorescences were much variable that the thrips captures in the aforementioned work. For instance, in the inflorescences, captures varied in samplings from 105 to 1,525 thrips adults per inflorescence. These high variable numbers lead to high variances yielding a low correlation.

## Conclusions

With an average of 3,138 *Frankliniella* thrips per inflorescence, the mango in Chiapas possibly has the highest numbers of thrips reported for mango in the world. Such large numbers of thrips can damage inflorescences rapidly, affecting fruit set. The use of colored sticky traps would be a good option for monitoring mango thrips in earlier stages of infestation to implement management tactics and avoid the building-up of thrips populations.

This study suggests the use of two different color traps for the main phytophagous thrips of mango. The white trap for the *Frankliniella* species, that also shows the least detrimental impact on natural enemies; and the yellow trap for *Scirtothrips*, with low detrimental effects on insect pollinators, although high impact to natural enemies. Considering that *Frankliniella* species are anthophilous and abundant during mango flowering, while *Scirtothrips* species are more critical during the mango set fruit, the use of white traps would be more advisable for monitoring *Frankliniella* populations during mango flowering, and yellow traps for monitoring *Scirtothrips* after set fruit. Since herbivorous insects combine visual and chemical cues to locate plants [79], future research on mango thrips should focus on traps that combine the most attractive color with semiochemicals, to improve trapping efficiency. Increasing thrips captures by baited colored traps, could evolved in a mass-trapping device that deploying in sufficient numbers in the field, would surely reduce the damage to flowers by thrips.

## Supporting information

**S1 Table. Thrips captured on colored sticky traps.** Total numbers of thrips captured on colored sticky traps throughout the flowering period of Ataulfo mango. Figures in each sampling represent the thrips captured in six treatments (colors) with eight replicates each.
(DOCX)

**S2 Table. Thrips collected from mango inflorescences.** Total numbers of Frankliniella thrips collected from mango inflorescences in seven samplings throughout the flowering period of Ataulfo mango. Each figure in the last column represents the total number of thrips collected from 20 mango inflorescences (10 before traps and 10 after traps).
(DOCX)

**S3 Table. Natural enemies.** Absolute abundance of natural enemies (parasitoids and predators) on colored sticky traps in Ataulfo mango agroecosystems.
(DOCX)

**S4 Table. Mango insect pollinators.** Absolute abundance of mango insect pollinators captured with colored sticky traps in Ataulfo mango agroecosystems.
(DOCX)

**S5 Table. Thrips catches.** Means of *Frankliniella* thrips in seven samplings throughout the flowering period of Ataulfo mango in Chiapas, Mexico. Figures of each sampling represent the average specimens collected in 20 mango inflorescences.
(DOCX)

# Acknowledgments

We are especially grateful to Alondra Martínez-Pérez, Arturo Pedraza-García and Miler Aguilar-Alvaro for their technical assistance and Eduardo R. Chamé-Vázquez for taxonomic identification of some beneficial arthropods reported in this paper. We are grateful to Jorge Santiago-Blay for his valuable comments on this manuscript. Lucia Carrillo-Arámbula received a M.Sc. scholarship from El Consejo Nacional de Ciencia y Tecnología (CONACYT) of Mexico.

# Author Contributions

**Conceptualization:** Lucia Carrillo-Arámbula, Francisco Infante, Adriano Cavalleri, Jaime Gómez, José A. Ortiz.

**Data curation:** Lucia Carrillo-Arámbula, José A. Ortiz, Ben G. Fanson.

**Formal analysis:** Lucia Carrillo-Arámbula, Francisco Infante, Ben G. Fanson.

**Investigation:** Lucia Carrillo-Arámbula, Francisco Infante, Adriano Cavalleri, Jaime Gómez, José A. Ortiz, Francisco J. González.

**Methodology:** Lucia Carrillo-Arámbula, Francisco Infante, Jaime Gómez, José A. Ortiz, Francisco J. González.

**Project administration:** Francisco Infante.

**Resources:** Francisco Infante.

**Supervision:** Francisco Infante, Adriano Cavalleri, Jaime Gómez, José A. Ortiz.

**Validation:** Lucia Carrillo-Arámbula, Francisco Infante.

**Visualization:** Lucia Carrillo-Arámbula, Francisco Infante.

**Writing – original draft:** Francisco Infante.

**Writing – review & editing:** Lucia Carrillo-Arámbula, Francisco Infante, Adriano Cavalleri, José A. Ortiz, Ben G. Fanson, Francisco J. González.

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
