## [Decision Letter · Decision Letter 0]

28 Jun 2022

PONE-D-22-14742Colored sticky traps for monitoring phytophagous thrips (Thysanoptera) in mango agroecosystems, and their impact on beneficial insectsPLOS ONE

Dear Dr. Infante,

Thank you for submitting your manuscript to PLOS ONE. After careful consideration, we feel that it has merit but does not fully meet PLOS ONE’s publication criteria as it currently stands. Therefore, we invite you to submit a revised version of the manuscript that addresses the points raised during the review process.

We look forward to receiving your revised manuscript.

Kind regards,

Ramzi Mansour

Academic Editor

PLOS ONE

Journal Requirements:

Reviewers' comments:

Reviewer's Responses to Questions

**Comments to the Author**

1. Is the manuscript technically sound, and do the data support the conclusions?

Reviewer #1: Yes

Reviewer #2: Yes

2. Has the statistical analysis been performed appropriately and rigorously? 

Reviewer #1: Yes

Reviewer #2: Yes

3. Have the authors made all data underlying the findings in their manuscript fully available?

Reviewer #1: No

Reviewer #2: Yes

4. Is the manuscript presented in an intelligible fashion and written in standard English?

Reviewer #1: Yes

Reviewer #2: Yes

5. Review Comments to the Author

Reviewer #1: This paper compares catches of thrips and beneficial insects on sticky traps of six colors in a mango crop in Mexico. It also records the species of thrips adults in the mango inflorescences at the same time. The paper is easy to follow, well presented and competently analysed. An accurate record of the species composition of thrips in mango inflorescences in Mexico is of interest and has clearly involved a large amount of work.

However, there are several weaknesses with this paper.

1. Earlier research (e.g. the paper below) has shown that subtle changes of shade of a trap color can affect trap catch by a factor of as much as 10. Thus, one experiment could show yellow catching more than blue, but another experiment with a slightly different blue could show blue catching more than yellow. Experiments with only one shade of a color cannot be generalised to all traps of that color, although many authors have done this in the past leading to apparently conflicting results.

Brødsgaard HF (1989) Coloured sticky traps for Frankliniella occidentalis (Pergande) (Thysanoptera, Thripidae) in glasshouses. J Appl Ent 107:136-140

Conclusions from the research need to be specific to the experiment. For example, conclusions could state that the white traps caught more than the yellow traps, but should not conclude that white catches more than yellow, because that is not necessarily true for all whites and all yellows. Recommendations need to be cautious in view of the few shades (one per color) that were tested.

2. In view of the large effect of subtle changes in the color of a trap, it is important to have an accurate record of the color of the trap. The paper only records reflectances down to 400 nm, but omits the UV range down to at least 350 nm, which is important for thrips. I know that spectrophotometers that record down to 350 nm are hard to find, but the region from 350-400 nm is critical. Paper or card traps are more likely to reflect UV in this range than painted traps because paper or card often has a mineral coating that reflects UV. In addition, recent research has suggested an important role of the glue:

van Tol, R.W.H.M., Tom, J., Roher, M., Schreurs, A. & van Dooremalen, C. (2021) Haze of glue determines preference of western flower thrips (Frankliniella occidentalis) for yellow or blue traps. Scientific Reports 11, 6557.

Thus, simple comparisons of color without consideration of the surface (in this case an acetate sheet and glue) do not provide enough information. The Methods need to be clear about whether or not the reflectance measurements included the acetate sheet and the glue.

Some more detailed points are:

1. The paper studies thrips in mango in Mexico. It would be useful to specify “Mexico” in the abstract.

2. Line 29. This should probably read “attractiveness to” or “attraction of”, but not “attractiveness of”. The attractiveness of thrips was not investigated.

3. Lines 40-42. This is an apparent contradiction. If the numbers caught on sticky traps showed no correlation with the numbers in inflorescences it is not immediately apparent why traps would be of any use for monitoring. The lack of a correlation would suggest that they are useless for monitoring. The recommendation for monitoring needs to be justified.

4. Line 105. Please specify Mexico.

5. Line 117. Please clarify what is meant by “contact paper”. Is it self-adhesive material? Is the material actually paper or is it vinyl? Was it matt or glossy?

6. Line 131. When traps are only 5 cm apart, a large catch on one will decrease the catch on the others because they compete for the same flying thrips. The effect is that differences between colors are exaggerated by the design. This could be mentioned in the Discussion.

7. Line 167. Did this include the acetate and the glue or not? Please specify.

8. Line 173. The lower limit is given as 400 nm, but the curves go a bit lower in Fig. 2. Please either correct the lower limit or correct Fig. 2.

9. Line 207. Trap catches could follow a negative binomial, but could easily be other distributions, such as poisson or quasipoisson. Was the validity of the negative binomial fit checked in any way? A negative binomial cannot just be assumed.

10. Lines 214-216 and lines 252-265. This analysis by wavelength is simplistic and unconvincing. Purple is identified as having peak reflectance in the shortwave, but Fig. 2 shows it has its largest peak in the longwave, so cannot be considered as “a shorter wavelength color”. The statistical comparisons that are reported (P<0.001, P=0.03 etc.) appear to include the traps as replicates and so the analysis contains pseudoreplication. There are only six different colors present, so there are only six true replicates. Replicates of each color are pseudoreplcates because they are not replicates of independent colors. I would simply omit this analysis. It is not valid to draw such conclusions from only six colors.

11. Lines 422-425. An important issue here is that you only had 8 replicates, which is rather few. The authors of reference 25 had rather more than 8. You would need more replicates to detect an effect.

12. Line 444. A more convincing argument is needed to justify traps for monitoring. You have found that there is no significant correlation between numbers in the inflorescences and trap catch and you have also pointed out that there are more thrips in the inflorescences than on the traps (line 231). The evidence suggests that looking at thrips in inflorescences would be a more sensitive and more reliable approach to monitoring than using traps. The case seems to be stronger for mass trapping than for monitoring.

13. Lines 451-454. I think that this proposal is only likely to be valid for Mexico. The authors present no evidence that it could be valid for other countries with other species. There is no evidence that color preferences are properties of genera rather than species. Although the results here are presented for Frankliniella spp. And Scirtothrips spp., each probably reflects the responses of just one dominant species rather than a genus response.

14. Table 1. It would be useful here to report the total number of specimens identified to species in the subsamples for traps and also for inflorescences. This would show the relative sampling effort.

15. Table 3. All the data in this Table are included in Fig. 6, so Table 3 can be omitted.

16. Fig. 3. Please give clearer units for the “number of individuals”. Are these thrips per trap per 72 hours?

17. Fig. 4. Please give clearer units for the “thrips per traps”. Are these thrips per trap per 72 hours? If these are the same units as for Fig. 3, please use the same axis label.

18. Fig. 6. Please state or indicate which graph is for which color. Reliance simply on symbol color is no good for people who are color blind.

19. Supplementary files. I cannot see the raw data for all the trap counts.

Reviewer #2: The submitted article PLOSE-D-22-14742 titled “ Colored sticky traps for monitoring phytophagous thrips (Thysanoptera) in mango agrosystems and their impact on beneficial insects”, is subdivided in the following paragraphs : 1)Introduction,2) Material and Methods, 3) Results, 4) Discussion, 5) Conclusions and 6) Reference List. Paragrapf 1 provides sufficient background and includes all proper references which are listed in paragraph 7. In M & M (2) all the applied methods and techniques in each step of the research design are clearly described, even through some parts need to be clarified or improved in technical details . Please, see my notes/ suggestions in the attached word file of the text, at lines 153, 156, 164, 173, 178, 201 (two comments/suggestions).The last comment at line 201 regards the combination of morpho-taxonomic identification of the main pest species collected and belonging to Frankliniella and Scirtothrips genera, with molecular techniques (DNA barcoding and sequencing ). Considering that one of the main questions proposed in this study was “ can be evaluated the relationships of the thrips abudance on traps and that one observed in the inflorescences?”, the results (3) obtained showed that the relationships between the thrips density on traps and that one observed into inflorescences is positive but not significant. In this case, my suggestion is an Improvement of statistical analysis of the field data for the main pest thrips captured (i.e. F.invasor) on colored sticky traps and density observed in mango inflorescences, in order to establish a significant correlation between the two screens. The relationships request to improve sampling methods to reduce the sampling variability and increase the degree of correlation.

6. PLOS authors have the option to publish the peer review history of their article (what does this mean?). If published, this will include your full peer review and any attached files.

Reviewer #1: No

Reviewer #2: No

---

## [Author Response · Author response to Decision Letter 0]

12 Aug 2022

“Colored sticky traps for monitoring phytophagous thrips (Thysanoptera) in mango agroecosystems, and their impact on beneficial insects”

[PONE-D-22-14742]

Response to Reviewers

We appreciate the positive and constructive comments made by the reviewers. Below, please find our responses. 

Reviewer #1: This paper compares catches of thrips and beneficial insects on sticky traps of six colors in a mango crop in Mexico. It also records the species of thrips adults in the mango inflorescences at the same time. The paper is easy to follow, well presented and competently analysed. An accurate record of the species composition of thrips in mango inflorescences in Mexico is of interest and has clearly involved a large amount of work. However, there are several weaknesses with this paper.

Comment. Earlier research (e.g. the paper below) has shown that subtle changes of shade of a trap color can affect trap catch by a factor of as much as 10. Thus, one experiment could show yellow catching more than blue, but another experiment with a slightly different blue could show blue catching more than yellow. Experiments with only one shade of a color cannot be generalised to all traps of that color, although many authors have done this in the past leading to apparently conflicting results.

Brødsgaard HF (1989) Coloured sticky traps for Frankliniella occidentalis (Pergande) (Thysanoptera, Thripidae) in glasshouses. J Appl Ent 107:136-140.

Conclusions from the research need to be specific to the experiment. For example, conclusions could state that the white traps caught more than the yellow traps, but should not conclude that white catches more than yellow, because that is not necessarily true for all whites and all yellows. Recommendations need to be cautious in view of the few shades (one per color) that were tested.

Response: We have read the paper mentioned by the reviewer and certainly, changes in shade of the blue color caught more individuals of F. occidentalis in comparison to white and yellow treatments. However, in our opinion both papers had different objectives. Meanwhile the Brødsgaard’s paper describes glasshouse experiments to find the optimum blue color shade to attract F. occidentalis, our paper aimed to determine the attractiveness of mango thrips towards several colors, using only one shade per color. In the literature there are hundreds of studies on F. occidentalis and many of them are dealing with color sticky traps preferences. This situation allows to carry out more specific studies testing different shades of color traps, but is not the case for thrips associated with mango, where the first step was to find the optimum colour preferences for thrips. Maybe in the near future we will perform experiments involving several grades of color shades. 

Comment. In view of the large effect of subtle changes in the color of a trap, it is important to have an accurate record of the color of the trap. The paper only records reflectances down to 400 nm, but omits the UV range down to at least 350 nm, which is important for thrips. I know that spectrophotometers that record down to 350 nm are hard to find, but the region from 350-400 nm is critical. Paper or card traps are more likely to reflect UV in this range than painted traps because paper or card often has a mineral coating that reflects UV. In addition, recent research has suggested an important role of the glue:

van Tol, R.W.H.M., Tom, J., Roher, M., Schreurs, A. & van Dooremalen, C. (2021) Haze of glue determines preference of western flower thrips (Frankliniella occidentalis) for yellow or blue traps. Scientific Reports 11, 6557.

Thus, simple comparisons of color without consideration of the surface (in this case an acetate sheet and glue) do not provide enough information. The Methods need to be clear about whether or not the reflectance measurements included the acetate sheet and the glue.

Response: In methods we have corrected the record of the spectral range of reflectance that certainly went from 350-1000 nm. We have also clarified that the spectral reflectance was measured without the acetate and the glue. We are sorry in failing to measure the reflectance colors with the glue, but given that the spectrophotometer was located in another city, far away from our city where we performed experiments, this part of the study was especially difficult to carried out. Since we used a clear glue (not the whitish one) in experiments, variations in color reflectance (with and without glue) were expected to be very slight for having an influence on thrips catches. We agree that reflectance of colors could be imprecise when measured without the glue, but we do not believe this situation could change drastically our results on the thrips attraction to the different trap colors under field conditions.

Comment. The paper studies thrips in mango in Mexico. It would be useful to specify “Mexico” in the abstract.”

Response: This correction has been done.

Comment. Line 29. This should probably read “attractiveness to” or “attraction of”, but not “attractiveness of”. The attractiveness of thrips was not investigated.

Response: This correction has been done.

Comment. Lines 40-42. This is an apparent contradiction. If the numbers caught on sticky traps showed no correlation with the numbers in inflorescences it is not immediately apparent why traps would be of any use for monitoring. The lack of a correlation would suggest that they are useless for monitoring. The recommendation for monitoring needs to be justified.

Response: We have modified this phrase to clarify.

We would like to point out that is inaccurate to assume that the lack of correlation would suggest that sticky traps are useless for monitoring thrips. Monitoring insects is not restricted to estimating their relative abundance. The concept of monitoring in agriculture involves the understanding on the absence/presence of pest organisms usually considered as pests on the crops, the early identification and detection of those organisms, and the estimation of damage, among others. We note that in the scientific literature over 95% of studies dealing with color sticky traps fail to correlate the number of insects caught on traps with the number of insects present on plants. However, these studies still very useful for monitoring insects. 

Comment. Line 105. Please specify Mexico.

Response: This correction has been done.

Comment. Line 117. Please clarify what is meant by “contact paper”. Is it self-adhesive material? Is the material actually paper or is it vinyl? Was it matt or glossy?

Response: This correction has been done.

Comment. Line 131. When traps are only 5 cm apart, a large catch on one will decrease the catch on the others because they compete for the same flying thrips. The effect is that differences between colors are exaggerated by the design. This could be mentioned in the Discussion.

Response: Although the reviewer could be right in assuming interference of color treatments in catching flying insects, we did not find in the scientific literature any potential interference among color sticky traps when they are disposed at 5 cm apart or more distance. However, as it was mentioned in methodology, color traps were randomly deployed on the T-shape stake in each block to prevent this potentially effect. 

Comment. Line 167. Did this include the acetate and the glue or not? Please specify.

Response: We have clarified in the text that the acetate and the glue were not included.

Comment. Line 173. The lower limit is given as 400 nm, but the curves go a bit lower in Fig. 2. Please either correct the lower limit or correct Fig. 2.

Response: The text in methodology has been corrected. the spectral range of reflectance went from 350-1000 nm.

Comment. Line 207. Trap catches could follow a negative binomial, but could easily be other distributions, such as poisson or quasipoisson. Was the validity of the negative binomial fit checked in any way? A negative binomial cannot just be assumed.

Response: As noted in the statistical analysis section, we used Pearson residuals to assess model validity. In particular, we checked for biases in residuals (e.g., misfit of the model structure – residual vs predicted, outliers/high leverage) and excessive dispersion/variation (larger Pearson residuals). All checks indicated that a negative binomial fit the data.

As for not selecting a Poisson distribution, the Poisson is a special case of the negative binomial (NB) in which theta goes to infinity. If the underlying distribution was Poisson, the theta estimate of the NB would be large enough to approximate a Poisson. Only one model had a large enough theta in which a Poisson might be fitted and hence we took the slightly more conservative approach of using a NB distribution.

As for the Quasi-Poisson (QP), the NB and QP often give similar results, but the NB has the added advantage of having a ‘real’ likelihood and in the exponential family of distributions. Hence, it is much easier to model using a GLM framework.

Comment. Lines 214-216 and lines 252-265. This analysis by wavelength is simplistic and unconvincing. Purple is identified as having peak reflectance in the shortwave, but Fig. 2 shows it has its largest peak in the longwave, so cannot be considered as “a shorter wavelength color”. The statistical comparisons that are reported (P<0.001, P=0.03 etc.) appear to include the traps as replicates and so the analysis contains pseudoreplication. There are only six different colors present, so there are only six true replicates. Replicates of each color are pseudoreplicates because they are not replicates of independent colors. I would simply omit this analysis. It is not valid to draw such conclusions from only six colors.

Response: There are two separate issues here that seem to be getting conflated. First, we will address our statistical analysis. We argue that our stat approach is valid and does not include pseudoreplication given our design. Our assumption is that every T-stake has the exact same six colors and we are making only inference about these six colors (trap color is treated as a fixed effect). We included T-stake as a random effect in model to account for a possible T-stake correlation structure. We also include day as day may differ. Thus, we account for possible correlational structures in our model and hence pseudoreplication (in sensu Hurlbert 1984) is not an issue.

What the Reviewer seems to be indicating is that to make inference about a color trap, one needs to sample from that color distribution. We see this not a statistical issue per se, but more of an interpretation issue.

Comment. Lines 422-425. An important issue here is that you only had 8 replicates, which is rather few. The authors of reference 25 had rather more than 8. You would need more replicates to detect an effect.

Response: This point is valid and is a weakness for this component of the results. For methodological reasons we could not performed more replicates.

Comment. Line 444. A more convincing argument is needed to justify traps for monitoring. You have found that there is no significant correlation between numbers in the inflorescences and trap catch and you have also pointed out that there are more thrips in the inflorescences than on the traps (line 231). The evidence suggests that looking at thrips in inflorescences would be a more sensitive and more reliable approach to monitoring than using traps. The case seems to be stronger for mass trapping than for monitoring.

Response: This issue was raised earlier by the Reviewer and addressed there.

Comment. Lines 451-454. I think that this proposal is only likely to be valid for Mexico. The authors present no evidence that it could be valid for other countries with other species. There is no evidence that color preferences are properties of genera rather than species. Although the results here are presented for Frankliniella spp. And Scirtothrips spp., each probably reflects the responses of just one dominant species rather than a genus response.

Response: We do not know yet the extent of our results. The opinion of the Reviewer may be possibly correct. 

Comment. Table 1. It would be useful here to report the total number of specimens identified to species in the subsamples for traps and also for inflorescences. This would show the relative sampling effort.

Response: The purpose of this table is to show the diversity of thrips only. We feel that the Table is overloaded and we would not like to add more information. Quantitative data of subsamples are presented in other sections, either on the supplementary information or raw data link. 

Comment. Table 3. All the data in this Table are included in Fig. 6, so Table 3 can be omitted. Fig. 3. Please give clearer units for the “number of individuals”. Are these thrips per trap per 72 hours?

Response: The Table 3 has been deleted. Thanks!

Comment. Fig. 4. Please give clearer units for the “thrips per traps”. Are these thrips per trap per 72 hours? If these are the same units as for Fig. 3, please use the same axis label.

Response: This correction has been done.

Comment. Fig. 6. Please state or indicate which graph is for which color. Reliance simply on symbol color is no good for people who are color blind.

Response: This correction has been done.

Comment. Supplementary files. I cannot see the raw data for all the trap counts.

Response: A link for the raw data has been added: 

https://figshare.com/articles/dataset/Color_Sticky_Traps/20477715

Response to the comments made by the Reviewer #2 

on manuscript PONE-D-22-14742

Reviewer #2: The submitted article PLOSE-D-22-14742 titled “Colored sticky traps for monitoring phytophagous thrips (Thysanoptera) in mango agrosystems and their impact on beneficial insects”, is subdivided in the following paragraphs: 1) Introduction, 2) Material and Methods, 3) Results, 4) Discussion, 5) Conclusions and 6) Reference List. Paragraph 1 provides sufficient background and includes all proper references which are listed in paragraph 7. In M & M (2) all the applied methods and techniques in each step of the research design are clearly described, even through some parts need to be clarified or improved in technical details. Please, see my notes/ suggestions in the attached word file of the text, at lines 153, 156, 164, 173, 178, 201 (two comments/suggestions). The last comment at line 201 regards the combination of morpho-taxonomic identification of the main pest species collected and belonging to Frankliniella and Scirtothrips genera, with molecular techniques (DNA barcoding and sequencing ). Considering that one of the main questions proposed in this study was “ can be evaluated the relationships of the thrips abundance on traps and that one observed in the inflorescences?”, the results (3) obtained showed that the relationships between the thrips density on traps and that one observed into inflorescences is positive but not significant. In this case, my suggestion is an Improvement of statistical analysis of the field data for the main pest thrips captured (i.e. F.invasor) on colored sticky traps and density observed in mango inflorescences, in order to establish a significant correlation between the two screens. The relationships request to improve sampling methods to reduce the sampling variability and increase the degree of correlation.

Comment. This technique has to be better described considering also the times.

Response: This correction has been done. 

Comment. The AA. have to clarify if adults or larval specimens were mounted on permanent slides. Hoyer medium is not suitable for permanent adult thrips specimens –Canada Balsam is the best medium. In case of larval specimens, Hoyer medium can be used but larvae cannot be soaked in NaOH solution for four hours.....

Response: We have clarified this issue as requested. At the end of that paragraph we have added information regarding the method used for mounting specimens. 

Comment. I suggest the AA to record the main biological characteristics and feeding habits they selected in order to distinguish the two ecological groups.

Response: We have modified this paragraph and now the two ecological groups can be distinguished clearly.

Comment. I suggest the AA. to add more technical details in the use of spectrometer for the evaluation of reflectance of colored traps.

Response: This correction has been done. 

Comment. How the inflorescences were taken from plants? By hands or by cutting?

Response: They were collected by hand. The text has been corrected. 

Comment. I also suggest the AA. to combine the morpho-taxonomic identification of the phytophagous thrips species collected belonging to Frankliniella and Scirtothrips genera with molecular identification through DNA barcoding and sequencing. Such species (i.e. Frankliniella) are quite difficult to identify because the character states are variable and can produce misidentification.

Response: The Reviewer suggest using molecules techniques to help with the taxonomic identification of thrips. There are two issues with this suggestion: (i) we do not have access to equipment to carry out a molecular analysis and (ii) thrips samples where usually contaminated with glue and probably useless for DNA sequencing. Moreover, for some species, as Frankliniella invasor, etc., there are no sequences available in the major molecular databases for comparisons. We agree that some Frankliniella and Scirtothrips species ID are challenging, but the species we recorded are known to be common in mango orchards of Mexico and there is good literature to provide a reliable identification of the species listed in our work. Some authors of this study have been working on thips for over 10 years and the possibility of having a taxonomic misidentification would be rather rare. 

Comment. Delete this sentence because already recorded in M & M paragraph.

Response: This correction has been done. 

Comment. The assertion is a “weak trick”. Not suitable knowledge on biology of the three collected Frankliniella phytophagous species has been yet produced. I suggest to improve the statistical analysis only for the collecting data of F.invasor, the highest caught species. But also it needs demonstrate that species identification is correct.

Response: We have included in this paragraph some references that give support to the biology of Frankliniella. Regarding to the statistical analysis, we acknowledge the recommendation, but we think is better to include in the analysis all the Frankiniella species, as it was one of our main goals since the beginning of the study. Thanks!

---

## [Decision Letter · Decision Letter 1]

31 Aug 2022

PONE-D-22-14742R1Colored sticky traps for monitoring phytophagous thrips (Thysanoptera) in mango agroecosystems, and their impact on beneficial insectsPLOS ONE

Dear Dr. Infante,

Thank you for submitting your manuscript to PLOS ONE. After careful consideration, we feel that it has merit but does not fully meet PLOS ONE’s publication criteria as it currently stands. Therefore, we invite you to submit a revised version of the manuscript that addresses the points raised during the review process.

We look forward to receiving your revised manuscript.

Kind regards,

Ramzi Mansour

Academic Editor

PLOS ONE

Journal Requirements:

Reviewers' comments:

Reviewer's Responses to Questions

**Comments to the Author**

1. If the authors have adequately addressed your comments raised in a previous round of review and you feel that this manuscript is now acceptable for publication, you may indicate that here to bypass the “Comments to the Author” section, enter your conflict of interest statement in the “Confidential to Editor” section, and submit your "Accept" recommendation.

Reviewer #1: All comments have been addressed

Reviewer #2: All comments have been addressed

2. Is the manuscript technically sound, and do the data support the conclusions?

Reviewer #1: Yes

Reviewer #2: Yes

3. Has the statistical analysis been performed appropriately and rigorously? 

Reviewer #1: Yes

Reviewer #2: Yes

4. Have the authors made all data underlying the findings in their manuscript fully available?

Reviewer #1: Yes

Reviewer #2: Yes

5. Is the manuscript presented in an intelligible fashion and written in standard English?

Reviewer #1: Yes

Reviewer #2: Yes

6. Review Comments to the Author

Reviewer #1: I accept nearly all the responses of the authors.

I have a few minor comments that should be easy to fix.

I pointed out that the reflectance in the UV from 350-400 nm is very important for thrips responses. The authors say that the spectrophotometer readings did in fact go down to 350 nm. However, Fig. 2 appears to show readings down to only about 375 nm. If the data exist for wavelengths down to 350 nm, these should be included by extending the wavelength axis to show down to 350 nm. However, if the data for 350 nm are not shown because they are unreliable below 375 nm, it should be stated that measurements went down to 375 nm. At the moment it seems odd to state that the measurements were made at the critical wavelengths down to 350 nm but then to omit them from Fig. 2.

I am still not clear about the units for the axes labelled as “number of individuals” or “thrips per trap” (Figs. 3, 4, 6). Are these all catches per 72 hours or have catches from several sampling dates been combined? What is the timespan of these catches? The rate of catch is of interest. This could be clarified easily by adding this in the figure legend if it does not fit on the axis.

Line 222. The authors responded that the negative binomial distribution was assumed and assessed. It would reassure the statistically interested reader if it could be stated that the distribution was not just assessed but also confirmed or found to fit well. The fit could have been assessed and found to fit badly.

The paper repeatedly mentions monitoring and investigates whether trap catches correlate with relative abundance, which would allow abundance monitoring. The results showed no significant correlation between trap catch and abundance in inflorescences, but concludes that traps would be useful for monitoring. The authors state that traps would be “a good option for monitoring mango thrips in earlier stages of infestation”. What I think is confusing is that the term “monitoring” is used through most of the paper in the context of measuring abundance but then uses it, without being explicit, to mean monitoring for detection rather than monitoring for abundance. I think it would be clearer to the reader if detection was made explicit. For example, the above sentence could say something like “a good option for monitoring mango thrips to detect them at earlier stages of infestation”.

I remain dubious about the statistical test to show that the three colors with peaks in the shorter wavelengths caught more thrips than the three colors that did not have peaks in the shorter wavelengths. This appears to be an a posteriori hypothesis. In other words, it appears that the authors looked for wavelengths that occurred in the colours that caught more and then tested this statistically on the data that had just suggested the hypothesis. Such a hypothesis is interesting but the statistical test that has been used is for an a priori hypothesis. However, I think that there is enough information there for the reader to draw their own conclusions.

Reviewer #2: (No Response)

7. PLOS authors have the option to publish the peer review history of their article (what does this mean?). If published, this will include your full peer review and any attached files.

Reviewer #1: No

Reviewer #2: No

---

## [Author Response · Author response to Decision Letter 1]

12 Oct 2022

Response to Reviewers

We appreciate the positive and constructive comments made by the reviewers. Below, please find our responses. 

Reviewer #1: I accept nearly all the responses of the authors. I have a few minor comments that should be easy to fix.

I pointed out that the reflectance in the UV from 350-400 nm is very important for thrips responses. The authors say that the spectrophotometer readings did in fact go down to 350 nm. However, Fig. 2 appears to show readings down to only about 375 nm. If the data exist for wavelengths down to 350 nm, these should be included by extending the wavelength axis to show down to 350 nm. However, if the data for 350 nm are not shown because they are unreliable below 375 nm, it should be stated that measurements went down to 375 nm. At the moment it seems odd to state that the measurements were made at the critical wavelengths down to 350 nm but then to omit them from Fig. 2.

Response: Even though our spectrometer provided readings down to 350nm the readings between 350 and 375nm were noisy and not reliable, therefore the reliable range of 375nm to 1000nm was considered for this work. This issue has been clarified in the material and methods section. 

Reviewer #1: I am still not clear about the units for the axes labelled as “number of individuals” or “thrips per trap” (Figs. 3, 4, 6). Are these all catches per 72 hours or have catches from several sampling dates been combined? What is the timespan of these catches? The rate of catch is of interest. This could be clarified easily by adding this in the figure legend if it does not fit on the axis. 

Response: We have added the units for each axis in the figure legends to clarify this issue.

Reviewer #1: Line 222. The authors responded that the negative binomial distribution was assumed and assessed. It would reassure the statistically interested reader if it could be stated that the distribution was not just assessed but also confirmed or found to fit well. The fit could have been assessed and found to fit badly.

Response: We have revised and modified this phrase in the text as suggested.

Reviewer #1: The paper repeatedly mentions monitoring and investigates whether trap catches correlate with relative abundance, which would allow abundance monitoring. The results showed no significant correlation between trap catch and abundance in inflorescences, but concludes that traps would be useful for monitoring. The authors state that traps would be “a good option for monitoring mango thrips in earlier stages of infestation”. What I think is confusing is that the term “monitoring” is used through most of the paper in the context of measuring abundance but then uses it, without being explicit, to mean monitoring for detection rather than monitoring for abundance. I think it would be clearer to the reader if detection was made explicit. For example, the above sentence could say something like “a good option for monitoring mango thrips to detect them at earlier stages of infestation”

Response: The context of the word ‘monitoring’ has been certainly stated in lines 86-90. However, we have modified the phrase to satisfy the reviewer. 

Reviewer #1: I remain dubious about the statistical test to show that the three colors with peaks in the shorter wavelengths caught more thrips than the three colors that did not have peaks in the shorter wavelengths. This appears to be an a posteriori hypothesis. In other words, it appears that the authors looked for wavelengths that occurred in the colours that caught more and then tested this statistically on the data that had just suggested the hypothesis. Such a hypothesis is interesting but the statistical test that has been used is for an a priori hypothesis. However, I think that there is enough information there for the reader to draw their own conclusions.

Response: These are post-hoc linear contrasts which are routinely used with statistical models and we are doing nothing different here. We followed standard practice: 1) is the model better than null model; 2) if so, are any factors significant; 3) if so, run post hoc comparisons to understand those relationships (e.g. the whole field of LSM/EMMEANS). We can understand the reviewer's point that one can argue about how the p-value should be adjusted since we also ran separate pairwise comparisons. However, the conclusions are robust (very low p-value) and we provide the 95%CI to support the strength of the relationships. The reviewer suggests not including support for these statements, but we argue stating that the shorter wavelength differed without any stat support (e.g. estimate with uncertainty) is rather handwavy and not convincing. Even the reviewer says that the readers will make the conclusion themselves, so why not provide the readers with the estimate and a measure of the uncertainty in that estimate?

Reviewer #2: No corrections were suggested by this reviewer

Thanks!

---

## [Editor Report · Decision Letter 2]

17 Oct 2022

Colored sticky traps for monitoring phytophagous thrips (Thysanoptera) in mango agroecosystems, and their impact on beneficial insects

PONE-D-22-14742R2

Dear Dr. Infante,

We’re pleased to inform you that your manuscript has been judged scientifically suitable for publication and will be formally accepted for publication once it meets all outstanding technical requirements.

Kind regards,

Ramzi Mansour

Academic Editor

PLOS ONE

---

## [Editor Report · Acceptance letter]

26 Oct 2022

PONE-D-22-14742R2 

Colored sticky traps for monitoring phytophagous thrips (Thysanoptera) in mango agroecosystems, and their impact on beneficial insects 

Dear Dr. Infante:

I'm pleased to inform you that your manuscript has been deemed suitable for publication in PLOS ONE. Congratulations! Your manuscript is now with our production department. 

Kind regards, 

on behalf of

Dr. Ramzi Mansour 

Academic Editor

PLOS ONE